# NHC-gold compounds mediate immune suppression through induction of AHR-TGFβ1 signalling in vitro and in scurfy mice

Xinlai Cheng [1,2,13]*, Stefanie Haeberle[3,13], Iart Luca Shytaj [4,5], Rodrigo.A. Gama-Brambila[1], Jannick Theobald[1], Shahrouz Ghafoory[1], Jessica Wölker[6,7], Uttara Basu[6,7], Claudia Schmidt[6,7], Annika Timm[6,7], Katerina Taškova[8,9,12], Andrea S. Bauer[10], Jörg Hoheisel[10], Nikolaos Tsopoulidis[3], Oliver T. Fackler [3], Andrea Savarino[11], Miguel A. Andrade-Navarro [8,9], Ingo Ott[6,7], Marina Lusic[4,5], Eva N. Hadaschik[3] & Stefan. Wölfl [1]

Gold compounds have a long history of use as immunosuppressants, but their precise mechanism of action is not completely understood. Using our recently developed liver-on-a-chip platform we now show that gold compounds containing planar *N*-heterocyclic carbene (NHC) ligands are potent ligands for the aryl hydrocarbon receptor (AHR). Further studies showed that the lead compound (MC3) activates TGFβ1 signaling and suppresses CD4[+] T-cell activation in vitro, in human and mouse T cells. Conversely, genetic knockdown or chemical inhibition of AHR activity or of TGFβ1-SMAD-mediated signaling offsets the MC3-mediated immunosuppression. In scurfy mice, a mouse model of human immuno-dysregulation polyendocrinopathy enteropathy X-linked syndrome, MC3 treatment reduced autoimmune phenotypes and extended lifespan from 24 to 58 days. Our findings suggest that the immunosuppressive activity of gold compounds can be improved by introducing planar NHC ligands to activate the AHR-associated immunosuppressive pathway, thus expanding their potential clinical application for autoimmune diseases.

[1] Institute of Pharmacy and Molecular Biotechnology, Heidelberg University, Im Neuenheimer Feld 364, D-69120 Heidelberg, Germany. [2] Buchmann Institute for Molecular Life Sciences, Goethe University Frankfurt, Max-von-Laue-Str. 15, D-60438 Frankfurt am Main, Germany. [3] Department of Dermatology, University Hospital Heidelberg, Im Neuenheimer Feld 440, 69120 Heidelberg, Germany. [4] Department of Infectious Diseases Integrative Virology, Heidelberg University, Heidelberg, Germany. [5] German Center for Infection Research (DZIF), Heidelberg, Germany. [6] Institute of Medicinal and Pharmaceutical Chemistry, Technische Universität Braunschweig, Beethovenstrasse 55, 38106 Braunschweig, Germany. [7] PVZ — Center of Pharmaceutical Engineering, Franz-Liszt-Straße 35A, 38106 Braunschweig, Germany. [8] Biozentrum I, Hans-Dieter-Hüsch-Weg 15, 55128 Mainz, Germany. [9] Faculty of Biology, Johannes Gutenberg Universität, Mainz, Germany. [10] Functional Genome Analysis, DKFZ, Heidelberg, Germany. [11] Department of Infectious and Immune-Mediated Diseases, Italian Institute of Health, Rome, Italy. [12] Present address: School of Computer Science, The University of Auckland, Auckland, New Zealand. [13] These authors contributed equally: Xinlai Cheng, Stefanie Haeberle. *email: x.cheng@uni-heidelberg.de

The aryl hydrocarbon receptor (AHR), a ligand-dependent transcription factor of the basic helix-loop-helix/Per-Arnt-Sim family, often acts as a sensor in response to environmental alternation[1]. The AHR gene is ubiquitously expressed in vertebrates and has an important role in development[2]. Its ligand-dependent activation is tightly controlled by a range of physiological responses[3]. The AHR was initially considered a mediator of the xenobiotic metabolism to detoxify dioxins and other toxic chemicals[3]. Because xenobiotic metabolism is part of the human defense system, the AHR is widely expressed in the immune system, including lymphocytes, macrophages and natural killer cells[4,5]. Compelling evidence has shown that ligand-mediated AHR activation triggers transforming growth factor beta (TGFβ) production, thereby, for instance, decreasing the expression of interleukin-2 (IL-2) and regulating the differentiation of CD4+ T cells to repress immune activity[6–9]. Velhoen et al. also reported that ligand-activated AHR contributed to the formation of Th17 T cells, a key mediator in the pathology of autoimmune diseases, implicating that the influence of AHR on the self-immune tolerance is largely related to the binding ligand[10]. So far, a large body of chemicals has been identified as endogenous or exogenous AHR ligands[2].

Autoimmune diseases, such as rheumatoid arthritis, inflammatory bowel disease, multiple sclerosis and type 1 diabetes mellitus, are characterized by over activity of the immune system, leading to tissue damage[11]. Immunodysregulation polyendocrinopathy enteropathy X-linked (IPEX) syndrome is a rare, heritable autoimmune lymphoproliferative disorder associated with severe diarrhea, diabetes, eczema, erythroderma, and liver diseases[12]. It is caused by the mutation of the FOXP3 gene and occurs in males in the early months of life. Immunosuppressive medications, like methotrexate (MTX), is used for the treatment[13]. However, at present bone marrow transplantation is the sole possible cure with many risks of complications; thus research is ongoing to identify and develop safer therapies for patients with IPEX[12]. In scurfy mice, a mouse model of human IPEX syndrome, treatment with TGFβ-differentiated FOXP3+ T cells from wt mice or autologous transplantation of hematopoietic stem cell containing a lentiviral vector for FOXP3 overexpression has been shown to sufficiently prevent skin inflammation[14,15]. Although the efficacy of gold compounds against rheumatoid arthritis was tested in the 20th century by Jacques Forestier[16], little is known about their effect on IPEX syndrome.

Recently, we established a liver-on-a-chip system consisting of a microfluidic chip equipped with two interconnected chambers to investigate cellular responses to liver-mediated drug metabolites[17,18]. Applying this system, we screened our own small molecules collection including some organometallics[19–21]. Interestingly, we found that N-heterocyclic carbene (NHC) gold complexes including our lead structure [di-(1,3-diethylbenzylimidazol-2-ylidene)]gold(I) iodide (MC3) (Supplementary Fig. 1A) and other gold organometallics are potent AHR ligands. In line with this effect, MC3 induced TGFβ1 and inhibited immune response in vitro and in vivo; however chemical and genetic inhibition of the AHR or TGFβ1 signaling pathways antagonized these effects. Thus, our data identify planar NHC gold organometallic compounds as activators of the AHR-TGFβ1 axis and show for the first time the potential of activating this signaling cascade as a complementary therapeutic target to enhance the immunosuppressive effect of gold compounds.

## Results

### Gold compounds induce CYP1A1 expression and are AHR ligands.

We recently developed a semi-automated, remotely controllable microfluidic chip system composed of two interconnected microfluidic chambers (Chamber 1 and 2, Supplementary Fig. 1B). The basic technical specifications of this system have been described previously[17] and are outlined in Supplementary Fig. 1B. We used hepatocellular carcinoma cell line HepG2 as a metabolite generator, which expresses the majority of human cytochromes P450 (CYPs) and is widely used in various microfluidic systems[17,18,22,23]. Thus, the compound of interest is metabolized in the chamber 1 (HepG2 cells) and cellular responses to metabolic products can be recorded in the chamber 2. In this study cellular viability was determined by the number of propidium iodide positive cells over total cells (Hoechst staining) and toxicity was calculated as the percentage of cleaved caspase 3. We screened our own small molecules collection (~ 200), including several organometallics we recently synthesized[21,24,–26]. The treatment of a gold complex bearing an NHC ligand MC3 (Supplementary Fig. 1A) led to significant differences in cell viability between chambers (Fig. 1b) and toxicity (Supplementary Fig. 1C). In addition, we studied the stability of MC3 and found that MC3 shows exceptional chemical stability in aqueous and polar environments at physiological temperatures over several days. Some decomposition occurred after extended periods (>24 h) in highly apolar media (Supplementary Fig. 1D), suggesting that this detoxification was associated with hepatocyte-mediated metabolism.

A number of exogenous and endogenous AHR ligands share planar structures with maximal dimensions of $14 Å \times 12 Å \times 5 Å$, including the prototype 2,3,7,8-tetrachlorodibenzo-p-dioxin (TCDD)[3]. Considering that MC3 carries a planar NHC ligand (Supplementary Fig. 1A), we hypothesized that MC3 can act as an AHR ligand and be metabolized by CYP1s. Indeed, global gene expression profiles obtained from DNA microarray analysis revealed that CYP1A1 was the top gene upregulated by MC3 among the ~ 20,000 genes tested, whereas other CYPs remained nearly unaffected (Fig. 1a). In addition, the KEAP1–NRF2 signalling cascade, which has been reported as a downstream pathway of AHR[27], was barely affected by MC3 treatment (Fig. 1a). MC3-mediated CYP1A1 expression was confirmed by RT-PCR, which showed ~ 100-fold induction of CYP1A1 after MC3 treatment (Fig. 1b). This induction was evident as early as 30 min after MC3 treatment and increased over time in a concentration-dependent manner (Supplementary Fig. 1E and 1F). Of note, 1 h MC3 treatment induced higher CYP1A1 expression than the prototypic AHR agonist TCDD (Supplementary Fig. 1G). To characterize the structural features responsible for the upregulation of CYP1A1 we tested two additional benzylimidazole-based, MC2 and MC4[19], as well as two imidazole-based NHC gold compounds, IO1 and IO2 (Supplementary Fig. 1H)[28,29]. We found that all NHC-gold compounds elevated CYP1A1 levels (Supplementary Fig. 1F), whereas hydrophilic metal-free salt, 1,3-diethylbenzimidazolium-miodide (MC1), did not. For comparison, we tested the effect of auranofin (Supplementary Fig. 1H), an FDA-approved gold complex for treatment of rheumatoid arthritis with covalent bonds to phosphine and thiol ligands in a linear arrangement. Results showed that auranofin is not an AHR ligand, because expression of CYP1s was not affected by its presence (Supplementary Fig. 1I). Taken together, our results indicate that MC2, MC3, and MC4 are activators of the CYP1 family most likely owing to the planar NHC structure.

Results obtained with immunostaining and immunoblotting showed that increased CYP1A1 protein expression after MC3 treatment is accompanied by the nuclear translocation of AHR (Supplementary Fig. 2A–2C, and Supplementary Fig. 6), an indicator of active AHR[30]. Co-incubation with resveratrol, a putative AHR antagonist[9], led to a significant reduction of both TCDD- and MC3-mediated CYP1A1 gene expression (Fig. 1c).

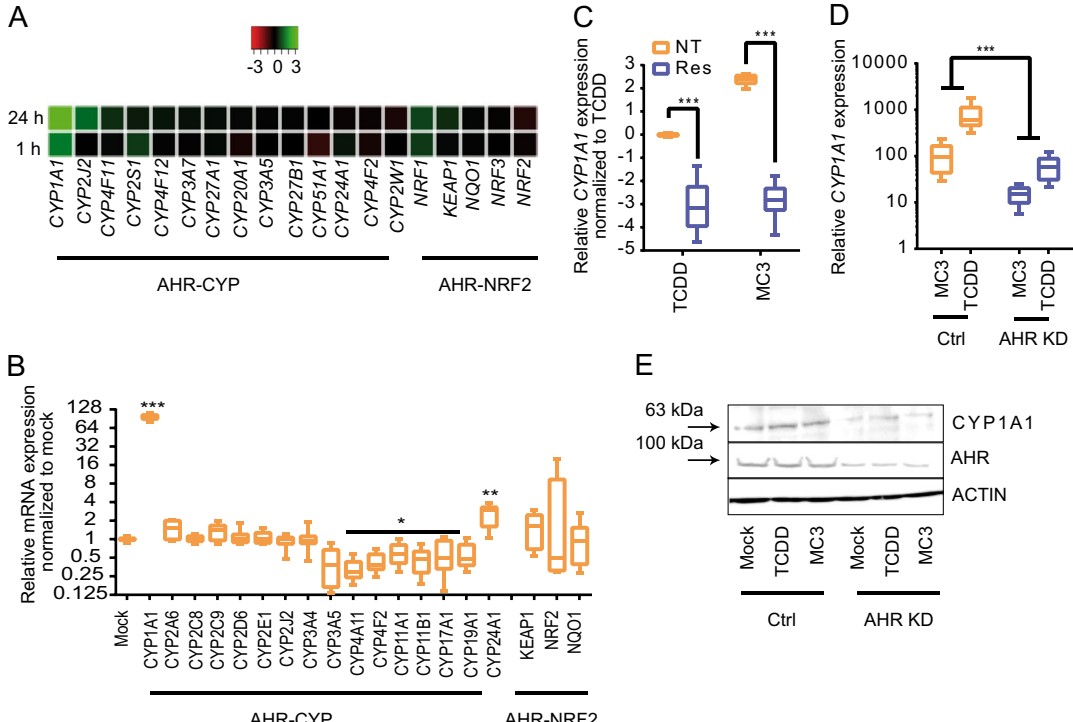

**Fig. 1 NHC gold complexes are potent AHR ligands. a** Heatmap of CYPs expression changes upon MC3 treatment for 1 h and 24 h from DNA microarray data (values are log of fold expression change versus DMF treatment). Expression profile of all members of the CYP 450 family and genes related to AHR-NRF2 signaling pathway. **b** Expression of CYPs and AHR-NRF2 related genes after MC3 treatment analyzed by RT-qPCR ($n \geq 6$). **c** AHR antagonist resveratrol (Res) significantly represses CYP1A1 expression in HepG2 cells treated with MC3 1 μM or TCDD 10 nM for 1 h. Data were normalized to the TCDD-induced CYP1A1 expression ($n = 9$). **d** Comparison of CYP1A1 expression in HepG2 AHR$^{KD}$ and CRISPR/CAS control cells, indicated as Ctrl. ($n = 9$). **e** Comparison of CYP1A1 and AHR expression in HepG2 control and AHR$^{KD}$ cells treated with TCDD 10 nM or MC3 1 μM for 24 h. One-way ANOVA $t$ test was performed. $*p < 0.05$, $**p < 0.01$, $***p < 0.001$;. lower and upper ends of the bars indicate the minimum and maximum values, respectively, and the centre represents the median. Error bars ± SD. The source data for 1**b–d** are provided as Supplementary Data 1.

We monitored the enzymatic activity of CYP1s in living cells using a commercially available dye Vividye BOMCC as described previously[18]. The result confirmed that resveratrol neutralized MC3-induced enzymatic activity of CYP1s (Supplementary Fig. 2D) and toxicity (Supplementary Fig. 2E).

With the help of CRISPR/CAS9 genome editing, we generated HepG2 AHR knockdown (KD, Supplementary Fig. 2F) cells and observed significant reductions of TCDD- and MC3-induced CYP1A1 expression at the transcription and translation levels (Figs. 1d and e and Supplementary Fig. 6). Together, our results demonstrate that MC3 and other NHC-containing gold compounds are potent AHR ligands.

**Immunosuppressive effect of MC3 in human primary T cells.** Previous findings have shown that hepatocytes are part of the defense system and express a large number of immune proteins to regulate T-cell activity[31]. In line with this, MC3 treatment in HepG2 cells influenced a significant number of genes related to immune response (Fig. 2a and Supplementary Fig. 3A), suggesting that MC3 modulates immune response through its interaction with the AHR.

To characterize direct immunomodulatory effects of MC3, we investigated different parameters of T-cell activation in freshly isolated human primary CD4$^+$ T cells treated with MC3. Experiments were conducted using a previously published protocol, in which resting CD4$^+$ T cells are incubated overnight with the drug and then activated by stimulating CD3–CD28 receptors[32,33]. We compared the effect of MC3 to those of

auranofin and TCDD. MTT assays first confirmed that at the optimized concentration MC3 (0.5 μM), auranofin (0.5 μM), and TCDD (10 nM) did not have a significant effect on the viability of resting CD4$^+$ T cells (Supplementary Fig. 3B). To assess the effect of each drug treatment on T-cell antigen receptor (TCR) proximal signaling events, we analyzed the ability to spread on a TCR-stimulatory surface and form circumferential F-ACTIN rich rings (Fig. 2b). This system mirrors key events of TCR engagement by antigen presenting cells and the observed actin polymerization is essential for downstream TCR signaling[34]. Although T-cell spreading and F-ACTIN ring formation were observed upon treatment with TCDD or auranofin, MC3 prevented cell spreading and significantly reduced ACTIN polymerization in response to TCR engagement (Fig. 2b). This defect in proximal TCR signaling translated in reduced downstream signaling because MC3 treatment led to significantly decreased cell surface levels of the T-cell activation marker CD25 (Fig. 2c and Supplementary Fig. 3C), and, importantly, intracellular production of the key cytokine IL-2 (Fig. 2d and Supplementary Fig. 3D) were significantly decreased upon treatment with MC3. TCDD did not show significant effects on T-cell activation, and, in line with previous results[35], treatment with auranofin reduced the levels of CD25, IL-2 and CD38 expression (Fig. 2c–4e and Supplementary Fig. 3C–3E). None of the drugs significantly modulated the expression levels of the CD3 co-receptor, suggesting that the decreased activation after MC3 treatment is not owing to altered availability of binding targets for the activating beads (Supplementary Fig. 3F and Supplementary Fig. 3G). Furthermore, impaired activation in MC3-treated

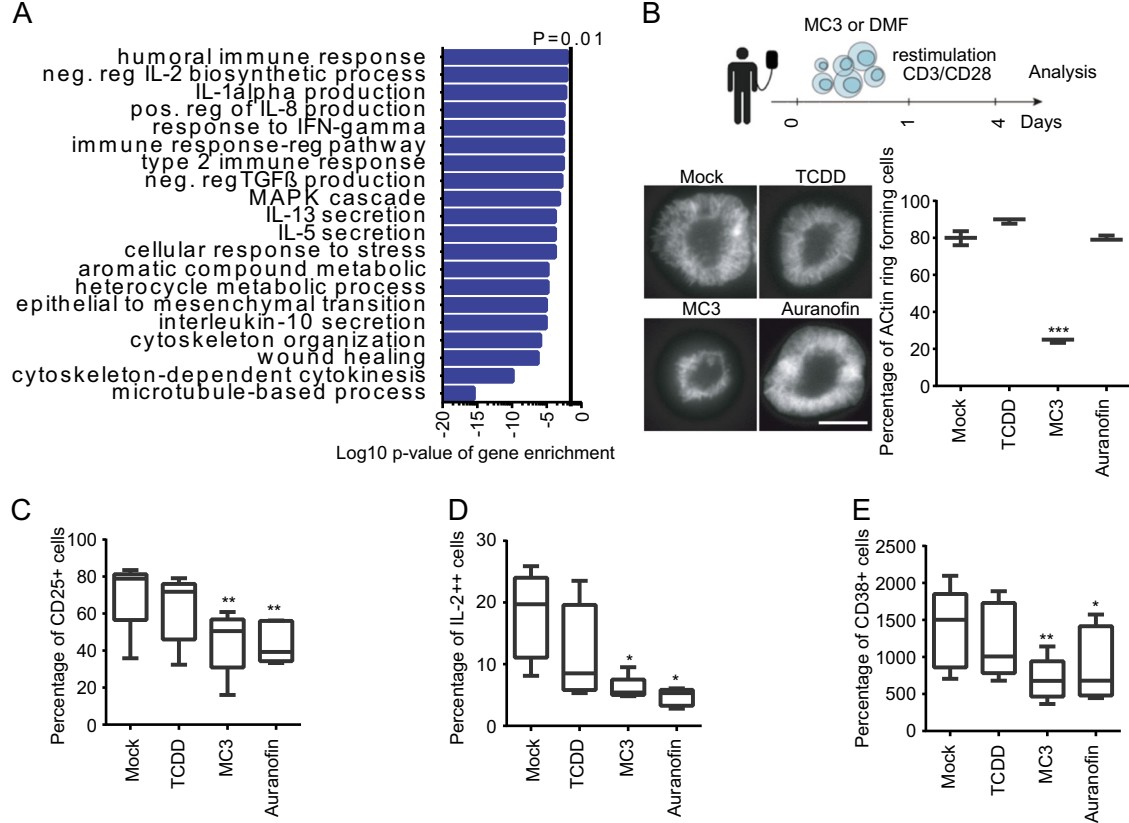

**Fig. 2 Immunosuppressive effect of MC3 in human primary T cells. a** Functional annotation analysis of DNA microarray data of RNA collected from HepG2 cells treated with 1 μM MC3 for 1 h and 24 h. **b–e** Inhibitory effect on primary CD4+ T-cell activation. Resting CD4+ T cells were left untreated or incubated overnight with 0.5 μM MC3, 0.5 μM auranofin or 10 nM TCDD. Cells were then activated through CD3-CD28 stimulation and assayed for: **b** ACTIN remodelling (formation of ACTIN rings) with the schematic representation of the treatment (Scale bar: 10 μm); **c** CD25 expression; **d** IL-2 production or **e** CD38 expression. For **b**, $n = 3$ donors, for **c–e**, $n = 5$ donors. Data were analyzed by one-way ANOVA or Friedman test. An appropriate transformation (Logit) was employed to restore normality when appropriate. One-way ANOVA $t$ test was performed. *$p < 0.05$, **$p < 0.01$, ***$p < 0.001$; lower and upper ends of the bars indicate the minimum and maximum values, respectively, and the centre represents the median. Error bars ± SD. The source data for 2**c–e** are provided as Supplementary Data 1.

lymphocytes was associated with decreased viability (Supplementary Fig. 3H), which is in line with previous reports[32,36]. Our results demonstrate that MC3 interferes with CD4+ T-cell activations and unlike auranofin, also affects TCR early signaling events, suggesting a non-overlapping effect of these drugs in different steps of T-cell activation.

**MC3-mediated immune suppression is TGFβ1-dependent.** Some potential mechanisms of action for the immunosuppressive effect of gold compounds such as thioredoxin reductase inhibition (TrxR) have been postulated; however, the precise molecular targets are still unknown[16,20]. As shown in Supplementary Fig. 4A, the result from the in vitro enzymatic activity assay revealed that MC3 is a much weaker TrxR inhibitor (<30% inhibition at 0.5 μM) than auranofin, which has an EC50 of 0.0005 μM[20]. Most likely, TrxR is not a primary target of MC3.

Early studies on primary T cells have highlighted the immuno-suppressive function of TGFβ1[37,38]. The AHR signaling pathway has been reported to regulate TGFβ1 production and modulate immune responses[5,6]. In good agreement, our global gene expression profile revealed that MC3-regulated signaling pathways were involved in TGFβ1 production, epithelial–mesenchymal transition and wound healing (Fig. 2a and Supplementary Fig. 3A). Thus, we investigated the role of TGFβ1 in MC3-mediated immune suppression in T cells. We screened a series of immortalized human T lymphocyte-related cells and found that TGFβ1 induction is accompanied by increased

CYP1A1 expression after MC3 treatment, except for Jurkat cells, which express a low level of AHR (Fig. 3a and Supplementary Fig. 4B)[39]. We compared the MC3-mediated gene induction of CYP1A1, TGFβ1, and IL-2 in SupT1 cells lacking AHR using either CRISPR/CAS9 (AHR KD) or siRNA (AHR siRNA). AHR deficiency clearly compensated the effect of MC3 on all four genes (Supplementary Fig. 4C).

Moreover, data obtained with an enzyme-linked immunosorbent assay confirmed the MC3-mediated increase of TGFβ1 only in SupT1 cells expressing AHR (Fig. 3b). Phosphorylation of SMAD3 is required for mediating TGFβ1 signaling[40], which was elevated by MC3 in SupT1 cells, but remained unaffected in AHR KD cells and in presence of the TGFβ1 inhibitor SB4 (Fig. 3c and Supplementary Fig. 4D and Supplementary Fig. 6). SMAD4 is required to transfer the cytosolic mediators of TGFβ1 signaling into nuclei and to trigger downstream expression of their target genes[40]. The treatment of either AHR siRNA or SMAD4 siRNA led to the inhibition of MC3-mediated SMAD3 phosphorylation, suggesting that AHR is an upstream of TGFβ/SMAD in this context (Fig. 3d and Supplementary Fig. 6). Furthermore, the cellular level of IL-2 expression was significantly reduced by MC3 in AHR-deficient SupT1 cells (Fig. 3e).

In addition to repress IL-2 expression, TGFβ1 can also induce FOXP3+ Treg cells to suppress immune activity[41]. Thus, we selected FOXP3 as the second indicator of TGFβ-related immunosuppression induced by MC3. Adding TGFβ1 to

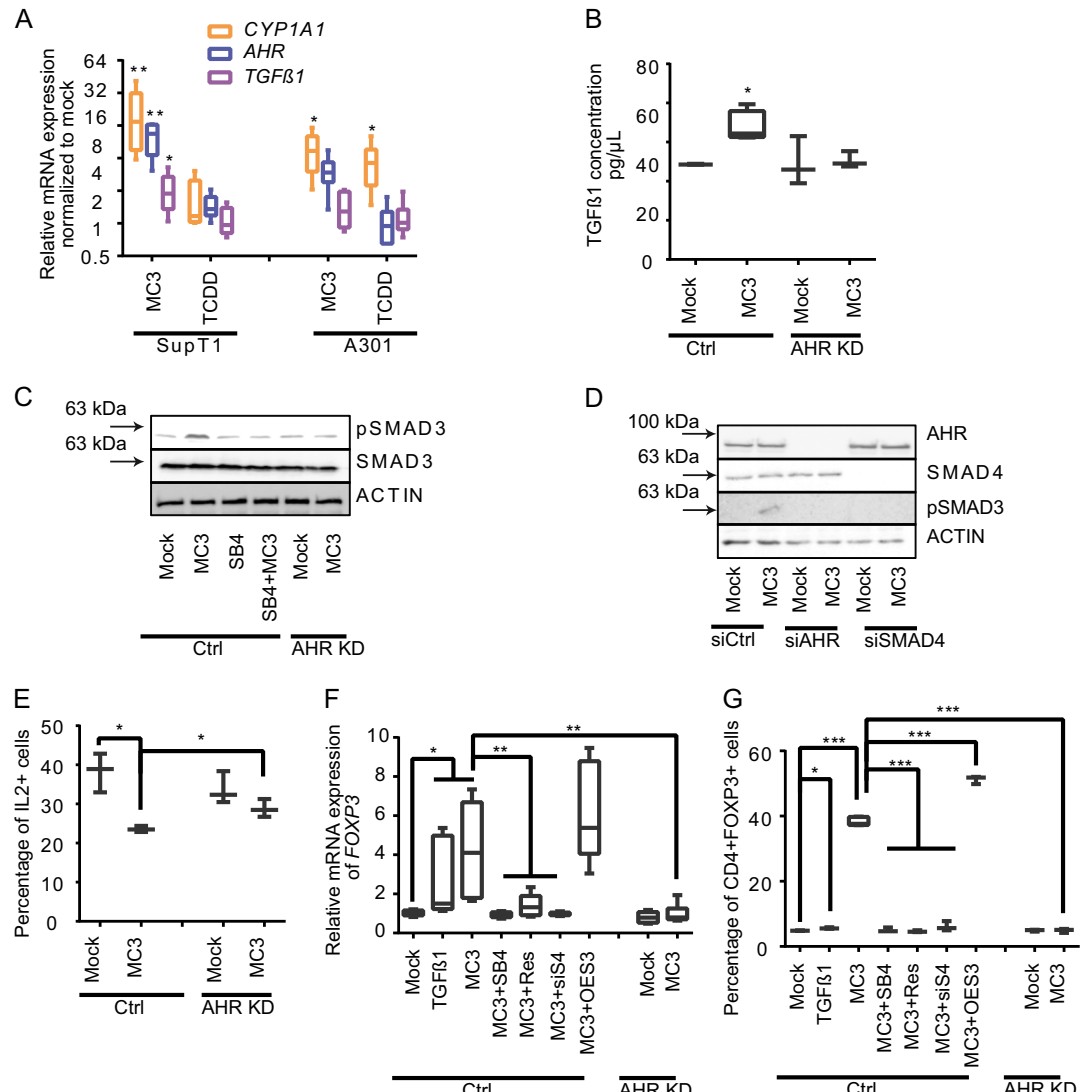

**Fig. 3 MC3-mediated immune suppression is TGFβ1 dependent. a** Expression of AHR-dependent gene in SupT1 and A301 cells treated with 0.5 μM MC3 for 24 h ($n \geq 6$). **b** Expression of TGFβ1 in cells treated with MC3 analyzed by ELISA ($n \geq 3$). **c, d** Immunoblotting analysis on the level of phosphorylation of Smad3 in SupT1 control, AHR KD cells, or AHR knockdown cells using siRNA or SMAD4 knockdown cells using. SB4: SB431542. **e** Comparison of IL2 expression cell population in SupT1 control and AHR KD cells treated with MC3 ($n = 3$). **f** Comparison of FOXP3 expression in SupT1 AHR KD cells and that in control cells after various indicated treatments ($n = 6$). **g** Comparison of the influence of various indicated treatments on the subpopulation of CD4$^{+}$FOXP3$^{+}$ in SupT1 AHR KD cells. MC3 (0.25 μM) was used for a 3-day treatment ($n \geq 3$). One-way ANOVA $t$ test was performed. *$p < 0.05$, **$p < 0.01$, ***$p < 0.001$; lower and upper ends of the bars indicate the minimum and maximum values, respectively, and the centre represents the median. Error bars ± SD. The source data for 3**a**, 3**b**, 3**e**–**g** are provided as Supplementary Data 1.

SupT1 cells promoted *FOXP3* gene expression (Fig. 3f) and increased the percentage of CD4$^{+}$FOXP3$^{+}$ cells (Figs. 3g and Supplementary Fig. 4E). As expected, MC3 treatment induced FOXP3 mRNA expression levels and increased the proportion of CD4$^{+}$FOXP3$^{+}$ cells (Fig. 3f and g). Either the chemical inhibition of AHR activity using resveratrol or the genetic knockdown of AHR compensated the MC3-induced FOXP3 expression. Moreover, the chemical inhibitor of the TGFβ1 receptor (SB4) or genetic knockdown SMAD4 (siS4) repressed the differentiation to FOXP3$^{+}$ cells, whereas overexpression of SMAD3 (OES3) further increased FOXP3 expression. Taken together, our in vitro results demonstrate that MC3 activates TGFβ1 to suppress immune activity at least by the reduction of IL-2 expression and/or activation of FOXP3$^{+}$ Treg.

**Enrichment of FOXP3$^{+}$ Treg in MC3-treated murine lymphocytes.** To confirm the involvement of TGFβ in murine

primary cells, we used a validated model of murine lymphocytes wherein TGFβ induction results in enrichment in Treg cells. In freshly isolated mouse CD4$^{+}$ T cells, MC3 increased the proportion of Treg cells (defined as CD4$^{+}$CD25$^{+}$FOXP3$^{+}$) from 4 to 14% at a concentration of 0.1 μM (Fig. 4a and Supplementary Fig. 5A–C). In addition, the expression of AHR-related genes such as *Cyp1a1* and *Cyp1a2* (but not *Cyp3a4*), as well as *Tgfβ1/2/3*, were elevated (Supplementary Fig. 5D). Moreover, resveratrol and SB4 neutralized the effect of MC3 (Fig. 4b). Thus, these results demonstrate that MC3 also repressed immune response in mouse CD4$^{+}$ T cells in an AHR-TGFβ-dependent manner.

**FOXP3-independent effects of MC3 in a mouse model of autoimmune disorders.** Promoting the differentiation of Treg cells from naive CD4$^{+}$ is one of the functions of TGFβ1 in immunity. Scurfy mice express a c-terminal truncated FOXP3 protein. This loss-of-function mutation leads to excessive

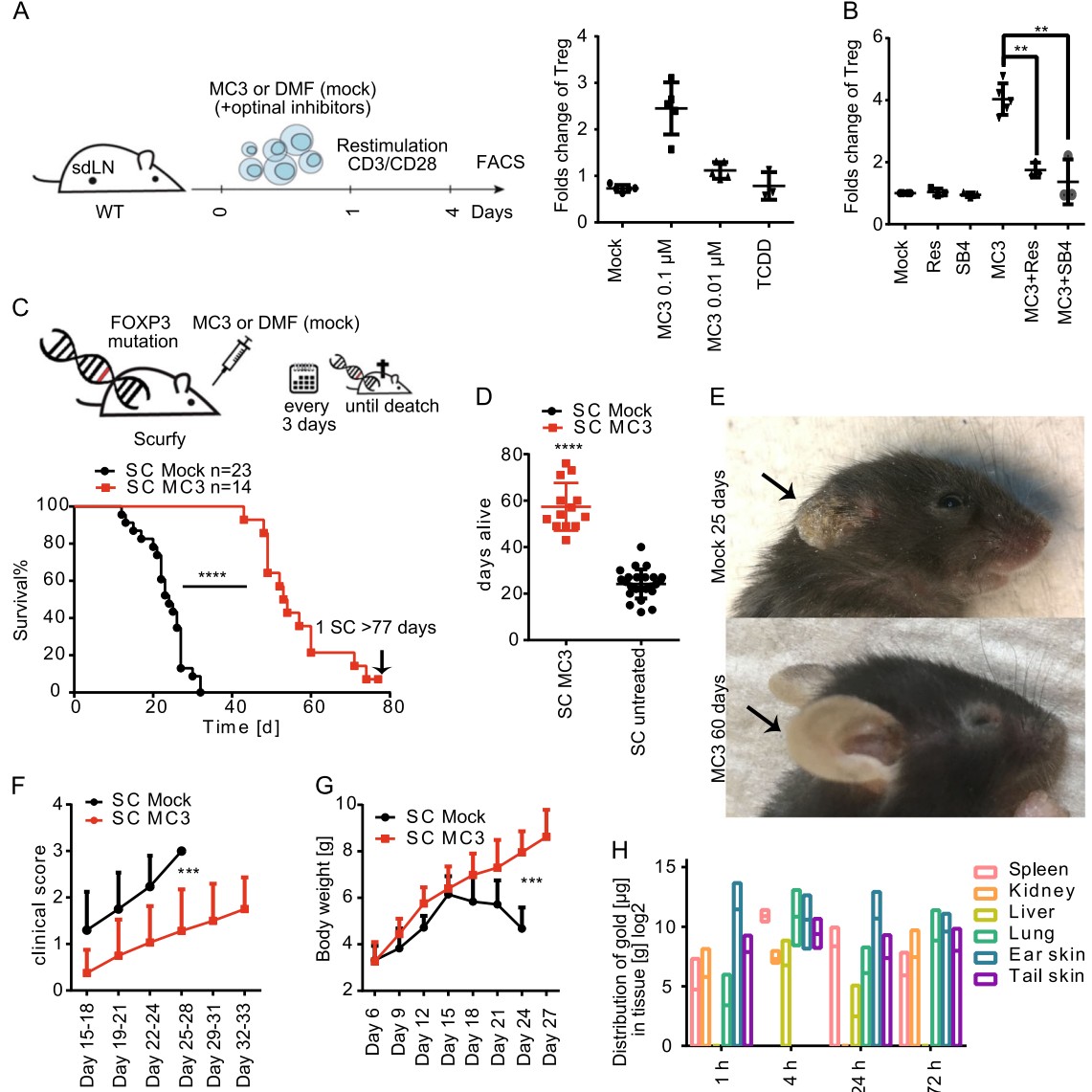

**Fig. 4 In vitro and in vivo immunosuppression of CD4$^+$ T cells by MC3. a** Fold changes of CD4$^+$CD25$^+$FOXP3$^+$ Treg-cells in the presence of MC3 or TCDD with schematic representation of the treatment ($n = 5$). **b** Antagonist effect of AHR inhibitor (Res, 10 μM) and TGFβ1 receptor inhibitor (SB4, 2.5 μM). $n \geq 3$ **c** Comparison of lifespan in scurfy mice (SC) treated with MC3 ($n = 14$) to non-treatment ($n = 23$) by survival ratio with schematic representation of survival experiment and **d** on average. **e** Comparison of autoimmune skin phenotypes of the ear in MC3-treated mouse (60 days) and those DMF-treated mice (25 days). **f** Comparison of clinical scores between DMF-treated and MC3-treated scurfy mice. The different grades of skin changes were scored as follows: No sign of skin inflammation is displayed by score 0, score 1 describes mild flaking of ear and tail skin, if flaking, erythema and thickening gets stronger and affects more areas a clinical score 2 is reached. Mice with strong skin manifestations, including open wounds, erosions and necrotic areas of ear and tail are defined by clinical score 3. Because of spontaneous death in DMF-treated scurfy mice, no control group is available after 29 days ($n = 6$). **g** Comparison of body weight between mock treatment and MC3 treatment in WT and SC mouse. Because of spontaneous death in DMF-treated scurfy mice, no control group is available after 29 days. **h** Biodistribution of gold in various tissues isolated from MC3-treated WT mice at the indicated point times. Statistical analysis was performed by different tests: **b** One-way ANOVA $t$ test was performed to compare more than two groups, for statistical analysis of survival ratios in C the Log-rank (Mantel–Cox) test was used. **d** Two-tailed unpaired $t$ test with Welch's correction and in **f**+**g** two-way ANOVA was performed *$p < 0.05$, **$p < 0.01$, ***$p < 0.001$, ****$p << 0.0001$. Lower and upper ends of the bars indicate the minimum and maximum values, respectively, and the centre represents the median. Error bars ± SD. The source data for 4**a–d**, 4**f–h** are provided as Supplementary Data 1.

expansion of autoreactive T helper cells and lethal multiorgan inflammation, all of which mimic the X-linked autoimmunity-allergic dysregulation syndrome in humans. As a result, sponta-neous death occurs between 16 and 25 days of age[42]. The effect of gold(I) compounds on the IPEX syndrome is still unknown. We studied the Treg-independent effects of MC3 in scurfy mice. Subcutaneous injection of MC3 every 3 days expanded the life-span of scurfy mice from an average of 24 days to 58 days with

some individual mice living up to 77 days (Figs. 4c and d). In addition, we observed a remarkable reduction of autoimmune skin manifestations, such as scales, crust, and erosions on ears and tails. In some cases, severe inflammation on the ears was com-pletely abolished even at 60 days (Fig. 4e). MC3-treated mice exhibited a delayed onset of skin manifestations and clearly lower clinical histopathology scores of the affected skin during the lifespan of N,N− dimethyl formamide (DMF)-treated scurfy mice

(Fig. 4f) and significantly greater body weight than DMF-treated mice (Fig. 4g). We also observed the abundant gold in spleen, kindey, ear skin, and tail skin detected by total reflection X-ray fluorescence spectrometer, which might explain the immunosuppressive effect of MC3 in scurfy mice (Fig. 4h).

Ectopic expression of activation markers, CD25 and CD69, was found in scurfy CD4$^+$ T cells[43]. Analysis of immunosuppressive markers showed that MC3 reduced these CD4$^+$ T-cell activation markers (CD25$^+$ in lymph node (LN) and CD69$^+$ in spleen) (Fig. 5a and b); however, the proliferation of MC3-treated CD4$^+$ T cells in scurfy mice was not affected (Supplementary Fig. 5E and S5F). In line with our previous results in vitro, MC3 induced the expression of *Tgfβ1* in the liver, thymus, and spleen as detected by RT-qPCR (Fig. 5c) and in situ hybridization (Fig. 5d). MTX is an immunosuppressant and has been used for the treatment of IPEX patients[13]. We isolated LN supernatant from scurfy mice treated with MTX or MC3. The quantified result using Legendplex clearly showed a significantly increased level of total TGFβ1 in the presence of MC3 (Fig. 5e). Furthermore, immunohistochemisty showed that MC3 significantly increased the expression of cytosolic and nucleic AHR in scurfy mice (Fig. 5f). This evidence supports the idea that MC3 is active in vivo as an immune suppressant and that, noteworthy, it may act through a Treg independent mechanism despite acting through TGFβ.

## Discussion
Our results highlight a novel mechanism of action for MC3-like (i.e., carriers of a planar NHC ligand) gold compounds exerted by potent AHR activation, leading to downstream TGFβ1 production and immune repression in vitro and in vivo (Fig. 5g). The therapeutic potential of this signaling cascade was demonstrated by the effects of MC3, which led to reduced systemic inflammation and organ damage in scurfy mice. Recently, Masiuk et al. reported that the injection of autologous hematopoietic stem cells with lentivirus-based overexpression of functional FOPX3 could rescue autoimmune phenotypes in scurfy mice; however, lifespan was not improved probably owing to the low efficacy of transplantation[15]. By contrast, our study demonstrated that MC3 treatment clearly prolongs the lifespan of scurfy mice. In scurfy mice, the effective concentration of MC3 is 0.16 mg/kg/3 days and we did not find any toxic effect, suggesting a good clinical tolerance for MC3.

The MC3 gold(I) complex was initially developed as an anticancer agent[19,20]. Our recently developed liver-on-a-chip microfluidic system enabled us to reveal that MC3 and other NHC-based organometallics are potent AHR agonists and effectively induce CYP1A1 expression. AHR is a ligand-activated transcription factor, which is of importance in xenobiotic metabolism and immune responses[1]. CYP1s are the major family of enzymes involved in AHR-mediated xenobiotic metabolism[44,45]. Computational scanning of human, mouse, and rat genes revealed the presence of dioxin response elements in the promoter regions of several thousand genes, including TGFβ1/2/3 and various ILs[46]. The interaction between the AHR and TGFβ1 signaling pathway has been reported previously[7,47]. Our data corroborate this interaction by showing that MC3-induced TGFβ1 expression does not occur in AHR-deficient cells or in the presence of an AHR inhibitor (Fig. 5g). Likewise, the immunosuppressive effects of MC3 disappeared in both cases. Moreover, several reports have indicated that structurally-distinct AHR ligands differently modulate AHR activity[9,10], which may account for the inability of TCDD to induce TGFβ1 and related immune repression in our experiments.

TGFβ1 is widely expressed in immune cells[48] and its deficiency causes multiple organ inflammation that leads to lethality in early mouse embryonic development[8], indicating the essential role of TGFβ1 in adaptive immune response. Early experiments showed that TGFβ1 blocks IL-2 transcriptional expression to inhibit T-cell proliferation[37,38]. In line with this, we observed decreased IL-2 production in SupT1 CD4$^+$ cells and primary CD4$^+$ T cells after MC3 treatment. In the presence of MC3, primary CD4$^+$ T cells were markedly less sensitive to exogenous stimulation as shown by the reduction of T-cell activation markers CD38 and CD25. In addition, the formation of immunological synapses was impaired as evidenced by the disruption of F-ACTIN rearrangements, which is one primary early steps of CD4$^+$ T-cell activation[49]. Notably, both MC3 and auranofin, impaired CD4$^+$ T-cell activation, but their mechanisms of action only partially overlapped: the latter did not exert an effect on F-ACTIN-mediated polarization. Moreover, in line with the finding that TGFβ1 promotes CD4$^+$ T cells to Tregs by activating FOXP3 expression[9,50], we detected an increased level of CD4$^+$FOXP3$^+$ in human and mouse CD4$^+$ T cells treated with MC3. Because scurfy mice lack functional Tregs, this suggests that the observed effect of MC3 was associated with other TGFβ1-related immunosuppressive signaling pathways (Fig. 5g). In general, functional TGFβ1 binds to receptors and phosphorylates regulatory-SMAD2/3, which in turn forms a complex with common SMAD4 and activates its downstream gene expression[40]. Reducing SMAD4 expression or using a chemical inhibitor of the TGFβ1 receptor compensated the immunosuppressive effect of MC3, whereas SMAD3 overexpression enhanced the function of MC3, indicating that MC3-mediated immunosuppression depends on TGFβ1 activity (Fig. 5g).

We observed that all three tested NHC-gold(I) complexes, but not auranofin, are AHR agonists. It is likely that organometallics containing planar NHC ligand are AHR agonists and can be metabolized by the CYP1 family. However, this hypothesis needs to be confirmed by adding various NHC-containing organometallics. Our in vivo data showed FOXP3-independent immunosuppression of MC3. This effect, as well as its FOXP3-dependent effect, needs to be elucidated in other animal models. It would also be interesting to examine how the AHR signaling pathway compensates loss-of-FOXP3 induced autoimmune response. Taken together, our results reveal that gold compounds containing planar NHC ligands utilize an alternative signaling pathway, which could explain their immunosuppressive effect and may have potential therapeutic use in immunology and related clinical applications. The mechanism of action showed herein overlaps only partially with that of a clinically approved gold compound, namely auranofin, As the clinical testing of auranofin to reduce immune activation in multiple diseases is expanding[51], leaving open the possibility to combine different classes of gold compounds to enhance their immunosuppressive effect in vivo.

## Methods
**Chemicals**. [Triphenylphosphine-1,3-diethylbenzylimidazol-2-ylidene)]gold(I) iodide (MC4) was synthesized as described[19,20,52]. TCDD was purchased from Sigma Aldrich (Germany). Resveratrol and Auranofin were from Santa Cruz (Germany). Antibodies of AHR (Biomol, Germany, AP5533C, 1:1000) and CYP1A1 were obtained from Abgent (Biomol, Germany, F50940, 1:1000). ACTIN (SC-47778, 1:1000) and VINCULIN (SC-73614) were from Santa Cruz. SMAD4 (9515, 1:1000), pSMAD3 (9520), and SMAD3 (9523, 1:1000) were from Cell Signaling (NEB, Germany).

**Cell culture**. HepG2 and HeLa cells were cultured in Dulbecco's modified eagle medium, and A301 and SupT1 in Roswell Park Memorial Institute (RPMI) 1640 containing 10% fetal bovine serum (FBS) and 1% Pen/Strp under 5% CO$_2$ at 37 °C in a humidified atmosphere and treated with compounds as indicated in the text.

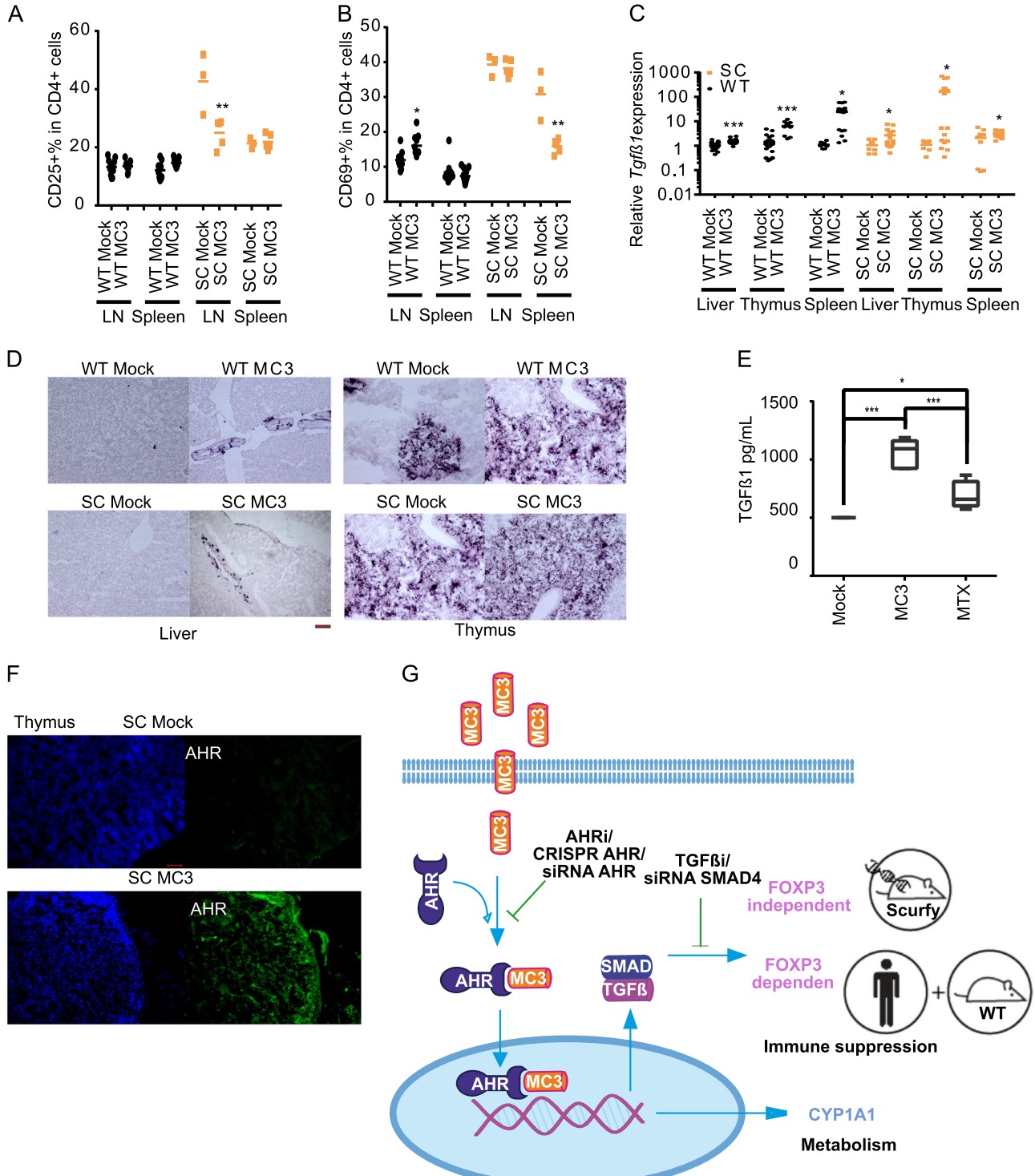

**Fig. 5 Characterization of MC3-mediated immunosuppression in scurfy mice. a** The percentage of CD4+CD25+ cells of the lymph nodes (LN) and spleen. **b** The percentage of CD4+CD69+ cells in the LN and spleen. **c** Relative expression of *Tgfβ1* in the liver, thymus, and spleen isolated from WT and SC mice treated with MC3. The data were normalized to respective mock treatment. **d** In situ hybridization of *Tgfβ1* expression in the liver and thymus isolated from WT and scurfy mice. Scale bar: 100 μM. **e** The quantification of secreted total TGFβ1 from PMA/IONO restimulated lymph node cells isolated from scurfy mice treated either with immunosuppressant methotrexate (MTX; $n = 5$) or MC3 ($n = 6$) or the mock (DMF, $n = 2$) control using Legendplex. **f** Immunohistochemistry analysis on the expression of AHR in non-treated and MC3-treated scurfy mice. Scale bar: 60 μM. **g** Summary of MC3-mediated immunosuppression in an AHR-TGFβ-dependent manner. The number of mice used for LD analysis $n_{WT\ mock} = 11$, $n_{WT\ MC3} = 11$, $n_{SC\ mock} = 3$, and $n_{SC\ MC3} = 5$; for spleen analysis $n_{WT\ mock} = 10$, $n_{WT\ MC3} = 13$, $n_{SC\ mock} = 3$ and $n_{SC\ MC3} = 5$. For Legendplex, $n_{SC\ mock} = 2$, $n_{SC\ MC3} = 6$, and $n_{SC\ MTX} = 5$. One-way ANOVA $t$ test was performed. *$p < 0.05$, **$p < 0.01$, *** $p < 0.001$. The source data for 5**a**, 5**b**, 5**c**, and 5**e** are provided as Supplementary Data 1.

Primary CD4+ T cells were isolated from total blood obtained from healthy donors in agreement with the regulations of the Institute for Clinical Transfusion Medicine and Cell Therapy (IKTZ, in Heidelberg) The isolation of primary human CD4+ T cells was performed by negative selection using the RosetteSep Human CD4+ T-Cell Enrichment Cocktail according to manufacturer's instructions. Murine CD4+ T cells were isolated from inguinal lymph nodes of WT mice via MACS separation using the negative selection isolation kit EasySep mouse CD4+ T Cell (Stemcell, Cologne, Germany) according to manufacturer's instructions. Isolated CD4+ T-cells were seeded at $10^6$ cells/mL in RPMI1640 containing 10% FBS and left untreated or treated with either 0.1–0.5 μM MC3, 0.5 μM auranofin, or 10 nM TCDD. Following overnight incubation, cells were either assayed by MTT and flow cytometry (resting cells) or activated by stimulating the CD3-CD28 receptors with Dynabeads Human T-Activator or Dynabeads Mouse T-Activator, respectively (Thermo Fisher Scientific, Darmstadt, Germany) according to manufacturer's instructions and left for 24–72 h in an incubator at 5% $CO_2$ and 37 °C. Activating beads were then magnetically removed and cells were used for flow cytometric analysis or for evaluating TCR-mediated actin ring formation.

To monitor proliferation ability after treatment with MC3, autanofin or TCDD, isolated murine CD4+ T cells were stained with CFSE (Thermo Scientific) for 20 min at 37 °C, followed by incubation with RPMI medium containing 10% FCS to stop the staining reaction.

siRNAs were purchased from (Thermo Scientific) and knockdown experiments were followed the previous work[25]. SMAD3 overexpression plasmid was purchased from Addgene (#14052) and performed as described before[53].

Mycoplasma contamination was determined routinely.

**Microfluidic device.** Cells were cultured in a chip consisting of two interconnected chambers. Each chamber comprises three inlet/outlet ports for cell seeding and culture medium supply, and a cell culture area surrounded by a guidance barrier. Prior to seeding, chips where coated with Collagen R 0.2% solution (Serva, Germany) for 45 min at 37 °C and thoroughly washed with PBS. Cell type specificity in each designated chamber was achieved by the chip design and the seeding procedure, blocking the three ports of the opposing chamber. After 4 h medium was carefully exchanged and cells were left overnight to fully adhere. For microfluidic studies a low pressure syringe pump (nemesis, CETONI, Germany) was used in combination with a 1.0 ml syringe H 1/4"–28 Tubing Connector (ILS, Germany). Cells were treated with an intermitted flow profile of 5 min flow at 100 μL/h and 55 mins of 0 μL/h for a period of 48 h.

**Cell apoptosis determination on chips.** Cells were treated with increasing concentrations of MC3 for 24 h and fixed with 4% PFA at room temperature for 15 min and blocked with blocking buffer (5% goat serum, 1% BSA and 0.3% Triton X-100 in PBS) for 30 min. After aspirating the blocking buffer, cells were incubated with cleaved Caspase3 antibody (1:200, Cell Signaling, NEB, Germany) for 1 h. The secondary antibody (Goat anti-rabbit Alexa Flor 594; Dianova, Germany) was added and incubated for 30 min. Hoechst 33342 (1 μg/mL in PBS) was used to visualize nuclei. Images were taken on BIOREVO fluorescence microscope (BZ9000, KEYENCE). Image J software (the particle count function) was used to calculate the percentage of viability, as we described previously[17].

**CRISPR/Cas9 knockdown AHR.** The CRISPR/Cas9 AHR plasmid (SC-400297-NIC-NIC) was purchased from Santa Cruz (Germany). The sequences of gRNA are CGGTCTCTATGCCGCTTGGA for strand A and TAATACAGAGTTGGACCG TT for strand B. Lipofectamine 3000 (Life Technologies, Germany) was used for transfection. According to previous work[53], the transfection was carried out in a 24-well plate with seeding density of 400,000 cells/well for HepG2 and 50,000 cells/well for SupT1. A mixture of 0.5 μg/well DNA and 1 μL Lipofectamine 3000 was added and incubated for 24 h. For the selection, 1 μg/mL puromycin was used. The established HepG2 AHR^KD and SupT1 AHR^KD cell lines were cultivated in respective medium containing 10% FCS, 1% PS, and 1 μg/mL puromycin for HepG2 and 3 μg/mL for SupT1 cells. The medium was changed two times per week.

**Sulforhodamine B assay (SRB assay) and (3-(4,5-Dimethylthiazol-2-yl)-2,5-diphenyltetrazolium bromide assay (MTT assay).** Effects on cell growth were determined in cell lines using the SRB assay and MTT assay at an initial cell density of 5000 cells/well in a 96-well plate as previously reported[54]. Cytotoxicity was determined as percent survival, determined by the number of treated over dimethyl sulfoxide cells. The comparable results were obtained from three independent experiments[55]. For primary cells, viability was determined using the CellTiter 96 Non-Radioactive Cell Proliferation MTT assay (Promega GmbH, Mannheim, Germany) according to manufacturer's instructions. In brief, $3 \times 10^5$ cells were washed and plated in 100 μL medium per well in a 96-well plate. Wells containing only medium were included to serve as a blank control. Per each well, 15 μL of dye solution were added and the plate was left for 3–4 h in an incubator at 5% $CO_2$ and 37 °C. The reaction was then stopped by adding 100 μL of Solubilization/Stop Solution per well and the absorbance values were recorded at a wavelength of 570 nm using an Infinite 200 PRO reader (Tecan Group, Männedorf, Switzerland).

After subtraction of the blank, absorbance values were expressed as percentage of controls.

**qRT-PCR.** Quantitative real-time reverse transcription-PCR was performed according to manufacturer's instructions (Lightcycle 96, Roch, Germany)[19]. In brief, total RNA was isolated from cells using RNeasy kit from Qiagen. cDNA was generated by reverse-transcription of equivalent quantities of RNA using Proto-Script First Strand cDNA synthesis kit from NEB. qRT-PCR was performed using SYBR Green PCR master mix (qPCRBIO SyGreen Mix Lo-Rox, Nippon Genetics, Germany). Primer sequences are listed in Supplementary Table 1. Actin was used as an endogenous control.

**Immunoblotting.** Cell extracts were homogenized in urea-lysis buffer (1 mM EDTA, 0.5% Triton X-100, 5 mM NaF, 6 M Urea, 1 mM Na₃VO₄, 10 μg/mL Pepstatin, 100 μM PMSF and 3 μg/mL Aprotinin in PBS) as previously described[55]. The immunoblot was detected by using ECL solution. In all, 40 μg of total protein was resolved on 10% SDS–PAGE gels and immunoblotted with specific antibodies. Primary antibodies were incubated at a 1:1000 dilution in blocking buffer with gentle agitation overnight at 4 °C. The results shown are representative of at least three independent experiments[55].

**Immunocytochemistry.** HepG2 cells were seeded in a Ø–12 mm cover slip coated with Geltrex (Life Technologies, Germany) at a density of 200,000 cells/well. After 24 h cells, were incubated with MC3 as described in the text, fixed with 4% PFA at RT for 15 min, and blocked with blocking buffer (5% goat serum, 1% BSA and 0.3% Triton X-100 in PBS) for 30 min. The blocking solution was aspirated and incubated with AhR and CYP1A1 antibodies, respectively. The secondary antibody (Goat anti-rabbit Alexa Flor 594; Dianova) was added and incubated for 30 min. Hoechst 33342 (1 μg/mL in PBS) was used to visualize nuclei. Alexa Fluor 488 phalloidin was used to visualize F-ACTIN. Images were taken on BIOREVO fluorescence microscope (BZ9000, KEYENCE)[56].

**Flow cytometry.** Staining of surface markers (CD3, CD25, and CD38) was performed as previously described[57]. In brief, $3 \times 10^5$ cells were collected, washed twice in PBS containing 5% FCS and incubated with anti-CD3 (FITC conjugated; BD Biosciences, NJ, USA) anti-CD25 (APC conjugated; BD Biosciences, NJ, USA) or anti CD38 (Alexa 488 conjugated, Biolegend, San Diego, CA, USA) antibodies for 30 min at 4 °C in the dark. Cells were then washed and resuspended in PBS containing 5% FCS. Staining of live cells was performed using the Fixable Viability Dye eFluor 450 (Affymetrix-Ebioscience, Santa Clara, California, USA according to manufacturer's instructions. Intracellular staining for IL-2 (APC conjugated; BD Biosciences) was performed according to manufacturer's instructions. In brief, $3 \times 10^5$ cells were collected and resuspended in 250 μL in Fixation/Permeabilization solution, incubated for 20 min at 4 °C in the dark, washed twice in BD Perm/Wash and incubated with an anti-IL-2 antibody for 20 min at 4 °C in the dark. Cells were then washed BD Perm/Wash and resuspended in PBS containing 5% FCS. Flow cytometric analyses were performed using a FACS Verse flow cytometer (BD Biosciences). Dot plots and histograms were generated with the FlowJo software (v 7.6.5).

**Gene expression analysis.** Gene expression profiling was conducted on the Illumina Human Sentrix-12v4 BeadChip array by the Genomics and Proteomics Core Facility of DKFZ. Gene expression analysis was performed in R 3.3.2 computing environment and packages from the open-source software development project Bioconductor 3.4[58]. Clustering and functional enrichment analysis was performed with the standalone tool STEM 1.3.11[59], whereas heatmap visualization of gene expression was performed with the package pheatmap 1.0.8. The genome-wide expression profiles of HepG2 cell line were determined using the Human-HT12-V4 Expression BeadChip. Three samples were profiled, corresponding to two MC3 treatments (profiled at 1 μM for 1 h and 24 h) and DMF-treated HepG2 (mock). Quality assessment and preprocessing of raw data were done with the package beadarray 2.24.2[60]. Prepossessing involved default image processing (with median-based local background), default Illumina removal for outlying observations, mean summarization of bead-level observations into probe-level data, quantile normalization of probe-level data, and log2 transformation. The resulting probes were matched to genes using annotations from the package illuminaHumanv4.db 1.26.0. After filtering probes that poorly matched the annotated genes (quality status of no-match and bad) as well as low-expression probes (detection score > 0.05), 13,010 probes were available for clustering analysis with STEM, a tool for clustering short time series data that can differentiate between real and random temporal expression patterns. For this purpose, we selected one probe per gene (the one with highest variance, if multiple probes matched to single gene) and converted the data into fold changes with respect to the not-treated HepG2. We analyzed the data as short time series of length three (corresponding to time point 0, 1, and 24 h after treatment). We applied STEM in default mode, except for the following settings: at least one measurement with absolute fold change of 0.65; significance of model profiles corrected by false discovery rate method; cellular component terms, terms with evidence code IEA and NAS were excluded from the Gene Ontology[61] enrichment analysis.

**Analysis of TVR-induced actin polymerization**. Primary human CD4+ T-cells were isolated from total blood in healthy donors (in agreement with the regulations of the Institute for Clinical Transfusion Medicine and Cell Therapy, IKTZ, in Heidelberg). The detailed protocol can be found in the supplementary information. TCR signaling induced circumferential actin polymerization was quantified as previously described[62]. In brief, stimulatory coverslips were prepared by coating microscope glass coverslips with a 0.01% poly-L-lysine solution for 10 min at RT. Subsequently poly-L-lysine was aspirated and the coverslips were coated with 10 μg/ml anti-CD3ε (clone HIT3a αCD3ε; BD Biosciences) in PBS and placed in a humidified chamber following 3 h incubation at 37 °C. The stimulatory coverslips were washed with PBS and 100 μl of drug treated T cells (1 × 10^6/100 μl) were loaded and allowed to adhere for 5 min before fixation with 3% PFA for 20 min at RT. The fixed cells were permeabilized for 2.5 min with 0.1% Triton X-100 and blocked with 3% BSA in PBS. Incubation with Phalloidin-Alexa-488 (1:1000 in PBS; Life Technologies) for 1 h was used to visualize F-actin. Samples were mounted on glass slides with Mowiol and analyzed by epifluorescence (Olympus IX81 S1F-3, cellM software). At least 100 cells per experiment and condition were analyzed for their ability to form a circumferential F-actin ring.

**Mice**. All animal experiments were approved with the animal proposal number G202-17 (Regierungspräsidium Karlsruhe, Baden-Württemberg, Germany). Scurfy Mice (B6.Cg-Foxp3sf/Y) were purchased from Jackson Laboratories (Bar Harbor, ME, USA). Female heterozygous scurfy mice were bred with C57BL/6 mice (WT) to generate male hemizygous offspring. Mice were held under specific pathogen-free condition in the Interfaculty Biomedical Facility (IBF) of University Heidelberg, Germany.

**In vivo experiments**. Scurfy or WT neonates were treated 3 days after birth with either 0.16 mg/kg MC3 or DMF as solvent control. 100 μl of MC3 or DMF diluted in medium was injected subcutaneously every 3 days. Body weight was controlled every 3 days. After 24 days, mice were killed and inguinal lns and spleen were used to generate single cell suspension, which was further analyzed by flow cytometry. The single cell suspension was stained with fixable dye Zombi Aqua (Biolegend), followed by staining with fluorescence-labeled antibodies against surface markers CD4 (RM4-4), CD25 (PC61), CD69 (H1.2F3), CD62L (MEL-14) in FACS buffer (PBS containing 1% FCS) for 30 min at 4 °C. For intracellular staining, the cells were fixed with fix-perm buffer (eBioscience) for 45 min at 4 °C. For nuclear staining, the transcription factor foxp3 (FJK-16s) was stained for 30 min at 4 °C in perm buffer. Analysis was performed with flow cytometer (Gallios, Beckman Coulter, Sinsheim, Germany). FACS data analysis was performed using the Kaluza software.

**In situ hybridization**. Paraffin blocks were prepared from different organs (spleen, thymus, kidney, and liver) and different treatments (in the same block for blind study) and cut into 4 μm slices, mounted on polyl-lysine-coated slides, air-dried O/N at 37 °C and stored at 4 °C (48 h).

In situ hybridization was performed as previously described[63]. mRNA extracted from C57BL/6 mouse brain, cDNA synthesized, *Tgfβ1* antisense RNA was generated (using forward 5′-CAGTGAATTGATTTAGGTGACACTATAGAAG TGACCATCGACA TGGAGCTGGTGAAAC-3′ and reverse 5′-CAGTGAATTG TAATACGACTCACTATAG GGAGACCTAAAGTCAA TGTACAGCTGCC GC-3′ primers).

**Definition of clinical score**. The different grades of skin changes were scored as follows: no sign of skin inflammation is displayed by score 0, score 1 describes mild flaking of ear and tail skin, if flaking, erythema, and thickening gets stronger and affects more areas a clinical score 2 is reached. Mice with strong skin manifestations, including open wounds, erosions, and necrotic areas of ear and tail are defined by clinical score 3.

**Statistics and reproducibility**. Data were analyzed with the GraphPad Prism software (v7.01, La Jolla, CA, USA). One-way analysis of variance (ANOVA) followed by Tukey's multiple comparisons test was employed when normal distribution could be assumed. Otherwise normal distribution was restored using the Logit transformation, followed by one-way ANOVA, or data were analyzed with an appropriate non parametric test, i.e., Friedman test followed by Dunn's multiple comparisons test.

All experiments have been independently repeated at least for three times, from which similar results were obtained.

**Reporting summary**. Further information on research design is available in the Nature Research Reporting Summary linked to this article.

## Data availability

DNA microarray can be obtained from GEO (Accession code: GSE125146). Supplementary Data 1 contains the source data underlying the graphs and charts presented in the article, including Figs. 1B, 1C, 1D, 2B, 2C, 2D, 2E, 3A, 3B, 3E, 3F, 3G, 4A, 4B, 4C, 4D, 4F, 4G, 4H, 5A, 5B, 5C, and 5E, as well as supplementary figs. 1G, 1I, 2E,3B, 3G, 3H, 4A, 4B, 4C, 4D, 5A, 5C, 5D, 5E, and 5F.

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

## Acknowledgements

We thank Bruker for lending us the TXRF spectrometer. This work supported by the DFG grant program (CH 1690/2-1) and the BMBF grant programs Drug-iPS (FKZ 0315398A-FKZ 0315398B) and SysToxChip (FKZ 031A303A-FKZ 031A303E). ILS acknowledges the post-doctoral fellowship provided by the Humboldt Foundation. J.W. acknowledges a Georg Christoph Lichtenberg fellowship by the State of Lower Saxony (graduate program µ-Props, "Processing of poorly soluble drugs at small scale"). M.L. and I.L.S. acknowledge the support of the German Center for Infection Research (DZIF), Heidelberg, Germany. This work was funded by the DFG-Transregio-Grant to S.H. and E.N.H. (TRR-156/C04).

## Author contributions

X.C. organized the study, designed and performed in vitro experiments, analyzed data, and wrote the manuscript. S.H performed mouse experiments, in vitro experiments on murine T cells, analyzed data, and helped to write the manuscript. I.L.S. performed experiments in human primary T cells, analyzed data, and wrote the manuscript. R.A.G.-B performed RT-qPCR. J.T. performed on-chips experiments. S.G. performed in situ hybridization experiment. J.W., C.S., and I.O. synthesized the compounds and studied the solution stability of MC3. U.B. and A.T. performed total reflection X-ray fluorescence spectrometer experiment. K.T. and M.A.A.-N. analyzed DNA microarray data. A.S.B. and J.H. performed DNA microarray analysis. N.T. conducted the F-ACTIN organiza-tion analyses. O.T.F., A.S., and M.L. supervised human sample experiments. E.N.H. supervised mouse experiments. S.W. participated in study design, organization of experiments, and writing of manuscript. All authors read the manuscript.

## Competing interests

X.C., I.O., and S.W. have filed a result-related patent application. All other authors declare no competing interests.
