## [Peer Review File · Communications Biology]

Reviewers' comments:

Reviewer #1 (Remarks to the Author):

The manuscript submitted to Commun. Biol. by X Chen et al. reports on the discovery of a potent gold-based immunosuppressant by inducing AHR-TGF β 1 signaling. The authors provide detailed mechanistic in vitro studies and an in vivo proof-of-principle. Especially, the normalized phenotypes in vivo and increased lifespan from 24 to 60 days in scurfy mice are remarkable when they are treated with the gold-drug.

The study is well designed, professionally performed and clearly presented. Although the novel aspects of this work evolve around a rather biomedical topic, the "scope of the journal includes all of the basic biological and biomedical sciences" and therefore the present study meets to scope of Commun. Biol. The authors may even emphasize more strongly that the lead compound MC3 follows a novel mechanism compared to the clinically established auranofin. This work provides an important impetus towards research in the field and broadens our understanding of the promising immunosuppressive effects of these gold-based AHR ligands. The manuscript may be accepted after considering the following points:

1. Describe the chip assay in some more detail. What is the actual read-out? Viability? How many compounds did the authors test?
2. The authors provided convincing results about the AHR-activation upon treatment with the gold-compound MC3. Is AHR activation always accompanied by immunosuppressive effects? Please comment. Moreover, it would be of interest to know whether Nrf2-KEAP1 is also activated upon treatment.
3. The authors compared their lead compound MC3 to its mono-carbene analogue and the ligand alone, as well as to a chemically and structurally very different gold-triphenylphosphane (and auranofin). Did other bis-NHC gold drugs also induce AHR activation?
4. Explain the "histopathology scoring parameters" of the in vivo experiment also in the main text.
5. It would be highly interesting to report the organ distribution of MC3 in treated mice, at least for the reported organs. This would add to the story since some effects on TGF β 1 were organ dependent, which may correlate with gold-uptake. It is also surprising that the in vivo control group consisted of around 20 mice, while the treated group seemed to consist of only 4 individuals? Please comment.
6. Add a scale for the fold-changes in Fig.S3.
7. Finally, since the authors isolated primary T cells, they should also provide an ethical approval number similarly to the in vivo experiments.

Reviewer #2 (Remarks to the Author):

In this study Xinlai and co-authors argue that N-heterocyclic carbene (NHC) gold complexes, specifically MC3, acts as an AHR ligand leading to TGF β 1 and immune suppression of aggressive autoimmunity. This is an interesting and provocative study but there are several concerns.

- 1) One of the major outcomes of the study is the increased survival of Scurfy mice. Unfortunately, the data, while provocative, appears to be limited to observations on 5 mice! Clinical score (Figure 4F)

does not show a difference between treated and untreated mice suggesting that the MC3 effect is simply to delay death. Whether this is a function of T regs is unclear. The study needs to be reproduced with greater numbers of MC3 treated mice in order to add power to the observations (e.g. is clinical score improved) and to provide material to determine if functional T regs are a plausible explanation.

2) The increased survival of Scurfy mice is presumably due to an increase in T regulatory cells. However, its unclear if this is due to a persistence of functional T reg numbers, a transient effect on functional T regs numbers, or an ability of MC3 to delay but not suppress other features of disease than can lead to death. There needs to be data on whether T regs are persisting out past 40-50 days and whether they are functional. Levels of AHR and TGF β 1 also need to be shown to be persisting to support the hypothesis of MC3 function.

3) MC3 appears to have a dramatic effect on ear inflammation (Figure 4E) even though these animals succumb to disease. What other features of clinical score are affected by MC3? The lack of data on clinical score past 33 days is presumably due to the small number of treated animals. A larger treatment group would allow better analysis of what features of clinical score are affected by MC3 treatment (see above). Additionally, how clinical score is determined should be described in the methods.

Reviewers' comments:

Reviewer #1 (Remarks to the Author):

The manuscript submitted to *Commun. Biol.* by X Chen et al. reports on the discovery of a potent gold-based immunosuppressant by inducing AHR-TGF β 1 signaling. The authors provide detailed mechanistic in vitro studies and an in vivo proof-of-principle. Especially, the normalized phenotypes in vivo and increased lifespan from 24 to 60 days in scurfy mice are remarkable when they are treated with the gold-drug.

The study is well designed, professionally performed and clearly presented. Although the novel aspects of this work evolve around a rather biomedical topic, the scope of the journal includes all of the basic biological and biomedical sciences and therefore the present study meets to scope of *Commun. Biol.* The authors may even emphasize more strongly that the lead compound MC3 follows a novel mechanism compared to the clinically established auranofin. This work provides an important impetus towards research in the field and broadens our understanding of the promising immunosuppressive effects of these gold-based AHR ligands. The manuscript may be accepted after considering the following points:

1. Describe the chip assay in some more detail. What is the actual read-out? Viability? How many compounds did the authors test?

In this study cellular viability was determined by the number of propidium iodide positive cells over total cells (Hoechst staining) and toxicity was calculated as the percentage of cleaved caspase 3 over Hoechst positive cells. Images (x20 magnification) were randomly taken on

BIOREVO fluorescence microscope (BZ9000, KEYENCE). Image J software (the particle count function) was used to calculate the percentage of viability, as we described previously (Theobald et al ACS Biomaterials Science & Engineering 2018). At the moment \approx 200 compounds have been investigated.

We added this information in our revision.

2. The authors provided convincing results about the AHR-activation upon treatment with the gold-compound MC3. Is AHR activation always accompanied by immunosuppressive effects? Please comment. Moreover, it would be of interest to know whether Nrf2-KEAP1 is also activated upon treatment.

The effect of AHR activation on immune system is dependent on the binding ligand. Both immune suppression and pro-inflammation have been reported (Quintana et al., 2008; Veldhoen et al., 2008). We cited this evidence in our original manuscript and have now emphasized it in the revision.

Regarding Nrf-KEAP1 signaling, we re-analyzed our gene expression data and found that KEAP1-NEF2 signaling was nearly not affected by MC3 (Fig. 1A), which was confirmed by qPCR (Fig. 1B). We added these new results in the revision.

3. The authors compared their lead compound MC3 to its mono-carbene analogue and the ligand alone, as well as to a chemically and structurally very different gold-triphenylphosphane (and auranofin). Did other bis-NHC gold drugs also induce AHR activation?

Recently, several novel bis-NHC gold compounds have been reported (Schmidt et al., 2019; Schmidt et al., 2017). We selected two NHC-containing compounds (IO1 and IO2, SI. 1H) from them and found that both promoted CYP1A1 expression in HepG2 cells.

4. Explain the 1C;histopathology scoring parameters 1D; of the in vivo experiment also in the main text.

The skin of scurfy mice was monitored during the treatment and skin alterations were scored using a clinical score. The different grades of skin changes were as follows: No sign of skin inflammation is displayed by score 0, score 1 describes mild flaking of ear and tail skin, if flaking, erythema and thickening gets stronger and affects more areas, a clinical score 2 is reached. Mice with strong skin manifestations, including open wounds, erosions and necrotic areas of ear and tail are defined by clinical score 3.

We added the definition in the figure legend and experimental section.

5. It would be highly interesting to report the organ distribution of MC3 in treated mice, at least for the reported organs. This would add to the story since some effects on TGF β 1 were organ dependent, which may correlate with gold-uptake. It is also surprising that the in vivo control group consisted of around 20 mice, while the treated group seemed to consist of only 4 individuals? Please comment.

We employed total reflection X-ray fluorescence spectrometer (TXRF) to study the organ distribution of gold. In this case wt mice were used. We detected abundant gold in spleen, which is in good agreement to our finding that MC3 promoted TGF β expression, because spleen is the major organ to secrete TGF β reported by Flanders et al (Oncotarget 2016).

Regarding the lifespan experiment, scurfy mice develop an X-linked lymphoproliferative disease, similar to X-linked autoimmunity-allergic dysregulation syndrome (XLAAD) in humans. Male scurfy mice die within 16-25 days of age. Scurfy mice have been maintained by mating scurfy females to WT males. Thus the possibility to get male scurfy mice is around 25% within the overall progeny. In the original manuscript we got 6 scurfy mice (1 died at day 48, 2 at 49, 2 at 60 and 1 at 70) from total 29 mice (around 21%). We agree that the effect of MC3 on survival of scurfy mice should include more data points. Therefore, we increased the number of MC3 treated scurfy mice to a total of 14 and could confirm the stable and strong effect of MC3 on the life expectancy in otherwise heavily affected scurfy mice. The clinical score was monitored during the course of treatment and we observed in single mice a complete absence of skin inflammation during MC3 treatment, however the majority of scurfy mice treated with MC3 showed a delayed skin inflammation which finally ended up in a similar clinical score at the end of their prolonged life expectancy. Nevertheless, we think that the significant prolongation of the life expectancy in scurfy mice is a robust and strong effect induced by MC3 given the fact that Scurfy mice suffer from very strong and fatal skin autoimmune inflammation by constantly activated autoreactive T and B cells.

6. Add a scale for the fold-changes in Fig.S3.

We added the scale to indicate fold-changes of gene expression in this figure

7. Finally, since the authors isolated primary T cells, they should also provide an ethical approval number similarly to the in vivo experiments.

The related information can be found in the supplementary information in our original submission. To emphasize the ethical approval, it now appears in the main text in the revision.

Reviewer #2 (Remarks to the Author):

In this study Xinlai and co-authors argue that N-heterocyclic carbene (NHC) gold complexes, specifically MC3, acts as an AHR ligand leading to TGF β 1 and immune suppression of aggressive autoimmunity. This is an interesting and provocative study but there are several concerns.

1) One of the major outcomes of the study is the increased survival of Scurfy mice. Unfortunately, the data, while provocative, appears to be limited to observations on 5 mice! Clinical score (Figure 4F) does not show a difference between treated and untreated mice suggesting that the MC3 effect is simply to delay death. Whether this is a function of T regs is unclear. The study needs to be reproduced with greater numbers of MC3 treated mice in order to add power to the observations (e.g. is clinical score improved) and to provide material to determine if functional T regs are a plausible explanation.

We thank the reviewer for the question highlighting the importance of immunosuppression of MC3 in vivo. We agree that the effect of MC3 on survival of scurfy mice should include more data points. Therefore, we increased the number of MC3 treated scurfy mice to a total of 14 and could confirm the stable and strong effect of MC3 on the life expectancy in otherwise heavily affected scurfy mice. The clinical score was monitored during the course of treatment and we observed in single mice a complete absence of skin inflammation during MC3 treatment, however the majority of scurfy mice treated with MC3 show a delayed skin inflammation which finally ends up in a similar clinical score at the end of their prolonged life expectancy. Nevertheless, we think that the significant prolongation of the life expectancy in scurfy mice is a

robust and strong effect induced by MC3 given the fact that scurfy mice suffer from very strong and fatal skin autoimmune inflammation by constantly activated autoreactive T and B cells.

Our finding suggested that MC3-induced immune suppression depended on TGF β . We used FOPX3 and IL2 as two readouts to indicate the activity of TGF β related immunosuppression in *in vitro* experiments. To confirm that the immune suppressive effect of MC3 is related to TGF β , we used scurfy mice and found the abundant expression of TGF β and expansion of lifespan of scurfy mice after treatment. However, in scurfy mice the effect of MC3 treatment cannot be mediated by Treg function, since the *foxp3* mutation in scurfy mice does not allow the generation of thymic-derived or peripherally induced regulatory T cells. We think that the MC3-induced expression of TGF β generates an immunomodulatory milieu favoring immunosuppression in the periphery and the TGF β production depends on the AHR signaling because inhibition AHR activity (using chemical inhibitor or knockdown AHR) blocked TGF β activity in our cellular assays (Fig. 3B-3G and Fig. 4B)

2) The increased survival of Scurfy mice is presumably due to an increase in T regulatory cells. However, its unclear if this is due to a persistence of functional T reg numbers, a transient effect on functional T regs numbers, or an ability of MC3 to delay but not suppress other features of disease than can lead to death. There needs to be data on whether T regs are persisting out past 40-50 days and whether they are functional. Levels of AHR and TGF β 1 also need to be shown to be persisting to support the hypothesis of MC3 function.

We thank the reviewer for this comment.

First at all, we would like to emphasize that to examine the immune suppressive effect of TGF β we used IL2 and FOXP3 as indicators in *in vitro* assay.

We agree that the suppression in scurfy mice leads to the hypothesis that MC3 induces TGF β expression that rather acts in a suppressive manner on other immune cells than converting Tregs. However, for the generation of the data in the current manuscript we use two different experimental systems to show the mechanisms of immunosuppression of MC3 treatment: First in vitro data in murine WT cells with undisturbed FOXP3 function were upregulation of TGF β expression results in the higher frequency of Treg cells with an immunosuppressive function (Fig. 4A and 4B). On the other hand, we performed MC3 treatment in scurfy mice which cannot generate functional Treg as they have a mutation in FOXP3. Here, MC3 treatment in the scurfy mice leads to a strong and robust effect in dampening the autoimmune inflammation and substantially prolonging the life expectancy. This effect in scurfy mice therefore must be FOXP3-independent and we favor the view that this strong immunosuppressive effect of MC3 in scurfy mice is mediated by a strongly increased TGF β expression after MC3 treatment as shown by several different methods including qPCR (Fig. 5C), in-situ staining (Fig. 5D), and cytokine release assays of skin draining lymph node cells of MC3 or DMF treated scurfy mice (Fig. 5E, a new result we performed during the revision, which confirmed that MC3 treatment induced TGF β expression).

Unfortunately, the scurfy mouse model does not allow studying Treg because they are not able to generate functional Tregs.

3) MC3 appears to have a dramatic effect on ear inflammation (Figure 4E) even though these animals succumb to disease. What other features of clinical score are affected by MC3? The lack of data on clinical score past 33 days is presumably due to the small number of treated animals. A larger treatment group would allow better analysis of what features of clinical score are

affected by MC3 treatment (see above). Additionally, how clinical score is determined should be described in the methods.

We appreciate the comment of the reviewer and acknowledge that further time points of the clinical score beyond 33 days should be considered as well. Due to fact that untreated scurfy mice die around 24 days +/- 1 day (In general, all scurfy mice died between 16-25 days of age, <https://www.jax.org/strain/004088>). they are not any longer available for comparative analyses and therefore we focused on the difference in the skin manifestation during 15-33 days.

The data of the life expectancy show that skin inflammation is reduced and delayed but cannot be completely prevented by MC3 in a 3 day treatment schedule. We agree that we need to clarify our clinical score in this context. We added the definition of clinical score in the figure legend and in material methods.

Quintana, F. J., Basso, A. S., Iglesias, A. H., Korn, T., Farez, M. F., Bettelli, E., Caccamo, M., Oukka, M., and Weiner, H. L. (2008). Control of T(reg) and T(H)17 cell differentiation by the aryl hydrocarbon receptor. *Nature* *453*, 65-71.

Schmidt, C., Albrecht, L., Balasupramaniam, S., Misgeld, R., Karge, B., Bronstrup, M., Prokop, A., Baumann, K., Reichl, S., and Ott, I. (2019). A gold(i) biscarbene complex with improved activity as a TrxR inhibitor and cytotoxic drug: comparative studies with different gold metallodrugs. *Metallomics* *11*, 533-545.

Schmidt, C., Karge, B., Misgeld, R., Prokop, A., Bronstrup, M., and Ott, I. (2017). Biscarbene gold(i) complexes: structure-activity-relationships regarding antibacterial effects, cytotoxicity, TrxR inhibition and cellular bioavailability. *Medchemcomm* *8*, 1681-1689.

Veldhoen, M., Hirota, K., Westendorf, A. M., Buer, J., Dumoutier, L., Renauld, J. C., and Stockinger, B. (2008). The aryl hydrocarbon receptor links TH17-cell-mediated autoimmunity to environmental toxins. *Nature* *453*, 106-109.

REVIEWERS' COMMENTS:

Reviewer #1 (Remarks to the Author):

The authors addressed all the points raised by the referee to a satisfactory degree. The manuscript may now be suitable for publication.